# Responses of the Mushroom *Pleurotus ostreatus* under Different CO_2_ Concentration by Comparative Proteomic Analyses

**DOI:** 10.3390/jof8070652

**Published:** 2022-06-21

**Authors:** Rongmei Lin, Lujun Zhang, Xiuqing Yang, Qiaozhen Li, Chenxiao Zhang, Lizhong Guo, Hao Yu, Hailong Yu

**Affiliations:** 1National Engineering Research Center of Edible Fungi, Institute of Edible Fungi, Shanghai Academy of Agricultural Sciences, Shanghai 201403, China; rmlin@mail.hzau.edu.cn (R.L.); zhanglujun@saas.sh.cn (L.Z.); liqiaozhen@saas.sh.cn (Q.L.); 20202106021@stu.qau.edu.cn (C.Z.); 2Shandong Provincial Key Laboratory of Applied Mycology, School of Life Sciences, Qingdao Agricultural University, 700 Changcheng Road, Chengyang District, Qingdao 266109, China; yangxq@qau.edu.cn (X.Y.); 198701007@qau.edu.cn (L.G.); 3Hubei Insect Resources Utilization and Sustainable Pest Management Key Laboratory, College of Plant Science and Technology, Huazhong Agricultural University, Shizishan Street, Wuhan 430070, China; 4CAS Key Laboratory of Insect Developmental and Evolutionary Biology, CAS Center for Excellence in Molecular Plant Sciences, Institute of Plant Physiology and Ecology, Chinese Academy of Sciences, Shanghai 200031, China

**Keywords:** *P. ostreatus*, proteomics, carbon dioxide, fruiting body, edible mushroom

## Abstract

Background: *Pleurotus ostreatus* is a popular edible mushroom in East Asian markets. Research on the responses of *P. ostreatus* under different carbon dioxide concentrations is limited. Methods: Label-free LC-MS/MS quantitative proteomics analysis technique was adopted to obtain the protein expression profiles of *P. ostreatus* fruiting body pileus collected under different carbon dioxide concentrations. The Pearson correlation coefficient analysis and principal component analysis were performed to reveal the correlation among samples. The differentially expressed proteins (DEPs) were organized. Gene ontology analysis was performed to divide the DEPs into different metabolic processes and pathways. Results: The expansion of stipes was inhibited in the high CO_2_ group compared with that in the low CO_2_ group. There were 415 DEPs (131 up- and 284 down-regulated) in *P. ostreatus* PH11 treated with 1% CO_2_ concentration compared with *P. ostreatus* under atmospheric conditions. Proteins related to hydrolase activity, including several amidohydrolases and cell wall synthesis proteins, were highly expressed under high CO_2_ concentration. Most of the kinases and elongation factors were significantly down-regulated under high CO_2_ concentration. The results suggest that the metabolic regulation and development processes were inhibited under high CO_2_ concentrations. In addition, the sexual differentiation process protein Isp4 was inhibited under high CO_2_ concentrations, indicating that the sexual reproductive process was also inhibited under high CO_2_ concentrations, which is inconsistent with the small fruiting body pileus under high CO_2_ concentrations. Conclusions: This research reports the proteome analysis of commercially relevant edible fungi *P. ostreatus* under different carbon dioxide concentrations. This study deepens our understanding of the mechanism for CO_2_-induced morphological change in the *P. ostreatus* fruiting body, which will facilitate the artificial cultivation of edible mushrooms.

## 1. Introduction

Mushrooms are macrofungi with epigeous or hypogeous distinct fruiting bodies [1,2,3]. The aggregation of hyphae occurs in the early stage of fruiting body development [4], and fruiting body shape can be immediately affected by environmental factors, including nutrients, humidity, temperature, carbon dioxide concentration, gravity, and light [5]. Mushrooms sense light, gravity, and carbon dioxide concentration to develop a proper pileus to enable the effective diffusion of spores. Environmental factors for fruiting body induction (light, temperature, and nutrients as three examples), development (light, gravity, and carbon dioxide concentration as three examples), and maturation or senescence (during/after sporulation and after artificial harvesting) of mushroom-forming basidiomycetes were researched in previous studies, wherein carbon dioxide affects the development of mushrooms [6,7,8]. Carbon dioxide is an important trace gas in the Earth’s atmosphere. Variation in its concentration can affect the growth and survival of living organisms on the Earth.

Previous studies demonstrated that CO_2_ is transported through membranes, is sensed by organisms, and acts as a key signaling molecule to control growth, differentiation, virulence, biotic interactions, etc. [9,10,11,12]. In yeast, CO_2_ can be transported into the cells mainly by simple diffusion, is then converted to HCO^3−^, and maintains CO_2_/HCO^3−^ homeostasis by carbonic anhydrase [13]. CO_2_ can also contribute to morphology, mating, sporulation, phenotypic switching, and virulence processes of fungi via the adenylyl cyclase/cAMP pathway [14]. Lu et al. revealed a new regulatory mechanism of CO_2_ signaling in fungi hyphal development by reducing Ume6 phosphorylation and degradation [15].

CO_2_ concentration could affect the development of fungal fruiting bodies. In particular, the differentiation of pileus would be inhibited under high CO_2_ concentration. Edible fungi need to consume a large amount of oxygen and metabolize CO_2_ in the growth process [16,17,18,19]. The excessive CO_2_ in the mushroom shed can inhibit the growth of mushrooms, and even lead to CO_2_ poisoning, manifested as the premature aging of mushrooms, formation of mushroom shape, etc., which may be caused by CO_2_ accumulation generated in the growth process of mushrooms due to poor ventilation [20,21]. Excessive CO_2_ in mushroom shed poses a threat to the health of mushroom cultivation management personnel, and thus deserves close attention. The appropriate CO_2_ concentration could stimulate the differentiation of fruiting bodies, and excessive CO_2_ concentration could inhibit the growth of hypha [22]. In the fruiting body stage, edible fungi are more sensitive to CO_2_ concentration. When the sporocarp is formed, it has strong respiratory function and the demand for oxygen increases sharply. When the CO_2_ concentration reaches more than 0.1%, it produces a toxic effect on the sporocarp. When the concentration of CO_2_ reaches 5%, it inhibits the differentiation of the pileus and even affects the formation of fruiting bodies. Therefore, the ventilation time should be reasonably determined according to the varieties of edible fungi and the growth period of various agents.

The Influence of the gaseous condition on fruiting body shape is relevant in commercial mushroom production. Respiration and the concentration of carbon dioxide during fruiting body formation have been investigated [6]. Respiration activity increases during primordia formation in the development of fruiting bodies, and a high concentration of carbon dioxide affects fruiting body morphology. Sensitivity to carbon dioxide has been investigated in the commercially cultivated mushroom varieties of *Flammulina velutipes*, *Pleurotus ostreatus*, *Pholiota nameko (microspora),* and *Lentinula edodes*; of these, *P. ostreatus* is one of the most sensitive to carbon dioxide [6,23]. In many mushroom species, the pileus is not fully developed and the stipe is spindly and elongated at a high carbon dioxide concentration. The morphology is similar to that produced in the absence of light. In the early stage of fruiting body development, sensitivity to carbon dioxide is more pronounced [6]. Elevated carbon dioxide affects the synthesis of the cell wall component R-glucan [7] and fruiting body cell morphology. The regulation of CO_2_ concentration in the cultivation environment is mainly achieved by setting the sealing time and ventilation time. However, the knowledge of the CO_2_ regulation mechanism in higher fungi is still unknown [5].

In the present study, we cultivated *P. ostreatus* at different concentrations of CO_2_ and the proteomes of the fruiting bodies were analyzed, which may help us to understand the mechanism underlining the fruiting body development of this mushroom.

## 2. Materials and Methods

### 2.1. Culture Conditions and Acquisition of the P. ostreatus Samples

*P. ostreatus* strain PH11 mycelia were cultured and subcultured in PDA media. For fruiting body production, the strain PH11 was inoculated into solid media in polypropylene bags (5 cm × 30 cm, 50 μm thickness). The formula of solid media was as follows: 55% cottonseed hull, 30% sawdust, 10% bran, 3% gypsum, 0.5% potassium dihydrogen phosphate, 0.5% urea, and 1% glucose, and the ratio of substrates to water was 1:1.5; the pH was natural. The cultivation bags with solid media were sterilized at 121 °C for 2 h, incubated at root temperature for 72 h, and sterilized at 120 °C for 2 h for the second time. Vegetative growth of *P. ostreatus* mycelia was performed at 25 °C with a humidity of 70% in darkness. After 40 days cultivation, the primordium was stimulated by water injection, low temperature, and light in a high-humidity environment. After the primordium formed, the high CO_2_ group bags were transferred into a CO_2_ chamber with 90–95% humidity, 12 h light/12 h dark, at 20 °C. Fruiting production at high CO_2_ concentration was carried out in a ZCLY-180ES CO_2_ chamber (Zhichu, Shanghai) with 90–95% humidity. The humidity and light were controlled by wireless humidifier and light. The control group bags were incubated in atmosphere with 90–95% humidity, 12 h light/12 h dark, and at 20 °C. The fruiting bodies were grown under different CO_2_ concentrations (1%, 0%) for 72 h, and the fruiting bodies were collected and stored at −80 °C for further analysis.

### 2.2. Protein Extraction and Peptide Digestion

Total proteins were extracted from the frozen *P. ostreatus* samples according to the following protocol: 100 mg frozen sample was taken into the centrifuge tubes and then 1 mL UT buffer (8 M urea, 0.1 M Tris-HCl pH 8.5) containing Thermo HALT protease and phosphatase inhibitor cocktail was added. The tissueLyser II was used to break the sample at 150 Hz for 60 s. The cell extract was treated by ultrasonication for 24 s (on for 6 s, off for 15 s). Tissue debris was removed by centrifugation (12,000× *g* for 10 min at 4 °C), and the supernatant was transferred into a new tube. Protein concentration was determined using the Micro BCA Protein Assay Kit (Thermo Fisher, Waltham, MA, USA). After adding 15 mg dithiothreitol (DTT), the sample was incubated at 37 °C for 1 h [24].

Afterwards, enzymolysis was performed according to the FASP method created by Wiśniewski et al. [25]. The extracted proteins (100 μg) dissolved in 300 μL UA buffer were taken into Pierce Protein Concentrators PES (10 K MWCO, 0.5 mL) (Thermo Fisher) to remove the low-molecular-weight impurities by centrifuging at 10,000× *g* for 30 min. A total of 50 mM of iodoacetamide was added to alkylate the proteins for 30 min at room temperature in the dark. The proteins were washed with 200 μL UA and 300 μL of 50 mM NH_4_HCO_3_ after removing the buffer by centrifugation. A total of 2 μg modified trypsin (Promega) in 100 μL of 50 mM NH_4_HCO_3_ was added into the ultrafiltration tube in a mass proportion of 1:50 (enzyme/protein). Enzymolysis was performed with gentle shaking at 37 °C for 12 h. After that, peptides were collected by centrifugation at 10,000× *g* for 15 min, and the residue peptides in the ultrafiltration tube were washed with 50 μL of 50 mM NH_4_HCO_3_ one more time. The salt in the pooled elutes was removed by using Merck Millipore ZipTip C18 resin (Darmstadt). Additionally, peptide concentration was measured by utilizing Pierce Quantitative Colorimetric Peptide Assay. The peptide sample was lyophilized on an RVC 2-25 CD plus vacuum concentrator (Christ), and stored at −80 °C for further analysis.

### 2.3. Label-Free LC-MS/MS Quantitative Proteomics Analysis

This analysis was performed with desalted peptides, which were reconstituted in 10 μL 0.1% formic acid, and was carried out using a Nano-LC system coupled with Orbitrap Fusion^TM^ Tribrid^TM^ (Thermo Fisher Scientific). The peptide sample (1 μL) was injected onto the Acclaim PepMap 100 nano trap column (75 μm × 2 cm, nanoViper 2PK C18, 3 μm, 100 Å) and then separated on an Acclaim PepMap^TM^ 100 analytical column (75 μm × 15 cm, nanoViper C18, 3 μm, 100 Å), and eluted in a 75 min nonlinear gradient program (mobile phase A: 0.1% (*w*/*w*) formic acid in water, mobile phase B: 0.1% (*w*/*w*) formic acid in 80% acetonitrile; 0–5 min, 4% to 8% B; 5–50 min, 8% to 20% B; 50–60 min, 20% to 30% B; 60–73 min, 30% to 90% B, 73–75 min, 95% B). The Orbitrap Fusion was operated in positive ion mode with spray voltage set at 2.2 KV and source temperature at 275 °C. The MS instrument was operated in data-dependent acquisition mode (DDA), with full MS scans over a mass range of *m*/*z* 350–1500 with detection in the Orbitrap (120 K resolution) and with auto gain control (AGC) set to 100,000 or a maximum ion injection time of 50 ms. For the survey scan, ten of the most abundance precursor ions with a charge state of 2+ to 6+ were selected for higher-energy collisional-dissociation (HCD) fragment analysis. The dynamic exclusion parameter was set at 60 s. For each group, four biological repeats were performed [24].

### 2.4. Peptides and Proteins Identification

The raw data were analyzed using Proteome Discoverer software suite Version 2.0 (Thermo Fisher Scientific) against the *P. ostreatus* PC15 proteome database from Uniprot (Proteome ID: UP000027073, accessed on January 2021) [26]. Protein identification was supported by at least two unique peptides with a false discovery rate lower than 0.05.

### 2.5. Bioinformatics Analyses

Raw data obtained from Proteome Discovery software were normalized. The *P. ostreatus* proteome was annotated and functionally enriched using the Gene Ontology tool (http://geneontology.org/, accessed on 26 April 2022) according to cellular component (CC), molecular function (MF), and biological process (BP). Gene Set Enrichment Analysis (GSEA) was used for interpreting gene expression data [27]. GO analysis was performed by using the R-ggcyto 1.24.0 tool [28].

## 3. Results

### 3.1. Phenotype of P. ostreatus under Different CO_2_ Concentrations

To investigate the effects of CO_2_ variations on the development of *P. ostreatus* PH11, young fruiting bodies of *P. ostreatus* PH11 were kept under a low concentration (atmospheric conditions) or high concentration (1%) of CO_2_. The fruiting body development was observed at both low and high CO_2_ conditions (Figure 1A–D); however, the stipes were much shorter in length (3.9 cm, 9.5 cm) and much thinner in the 1% carbon dioxide concentration (HCO) group as compared with the atmospheric condition (LCO) group (Figure 1B,E). A significant, quick pileus expansion at 48 h was recorded in the LCO group, whereas the pileus growth was significantly inhibited with no basidiospore shedding in the HCO group (Figure 1C,F). A difference in the pileus diameter was observed under low CO_2_ conditions compared with high CO_2_ conditions, suggesting a significant change in the function of the pileus due to the variation in CO_2_ conditions.

### 3.2. Proteomic Analysis of P. ostreatus Grown under Different CO_2_ Conditions

There are 56 unique proteins in HCO and 140 unique proteins in LCO. There were 1137 shared proteins in two different CO_2_ treatments according to the Venn diagram (Figure 2D). Principal component analysis shows that the four replicates of LCO (green) clustered more closely than HCO (red). Compared with the LCO group, the HCO group demonstrated a more divergent expression level. The second principal component varied greatly in HCO, especially HCO-4 and HCO-1. Pearson correlation shows that the correlation among different treatments was high with the lowest correlation coefficient of 0.93 (between HCO-4 and LCO-3), even in different carbon dioxide concentrations (Figure 2A,C, Appendix A). The protein expression profiles of the LCO group and HCO group displayed a higher correlation (>0.93) with each other. Furthermore, the principal component analysis showed similar results (Figure 2B). Additionally, Pearson correlation coefficient analysis and principal component analysis results reveal that the two groups of samples were reasonable with good correlations among biological repeats.

Compared with *P. ostreatus* under normal conditions, there were 131 differentially expressed proteins in *P. ostreatus* treated with 1% CO_2_ concentration (Figure 3 and Figure 4). Of 131 DEPs, 48 were uncharacterized proteins, which need to be annotated. A wide variety of DEPs were annotated with different biological functions. For instance, some DEPs were hydrolases (such as A0A067NG45 and A0A067NTT4), which catalyze the hydrolysis of a chemical bond. Ribosomal proteins, such as A0A067NG60 and A0A067N2P4, play roles in the formation and functioning of the ribosome. Chitinases, including A0A067N5C9 and A0A067NX05, play significant roles in stipe cell wall extension in mushrooms. Some DEPs are histones (including A0A067NL48 and A0A067P7W0), which provide structural support for a chromosome.

### 3.3. Gene Ontology (GO) Enrichment Analysis for DEPs

GO analysis for the DEPs to Molecular Function (MF), Cellular Component (CC), and Biological Process (BP) categories are as follows: Gene Ontology analysis was performed to classify the annotated DEPs. These profiled DEGs were categorized into three main GO categories, CC, MF, and BP. There were 131 DEPs in total. The 131 DEPs were divided into 22 terms, involving 4 BP, 6 CC, and 12 MF terms. The more conspicuous terms of each category were as follows: carbohydrate metabolic process (17), organic substance metabolic process (39), peptide metabolic process (8), and other metabolic processes (40) for BP; non-membrane-bounded organelle (13), intracellular non-membrane-bounded organelle (13), ribosome (7), intracellular anatomical structure (23), intracellular organelle (20), and organelle (20) for CC; and hydrolase activity (46), exopeptidase activity (7), hydrolase activity acting on hydrolyzing O-glycosyl compounds (14), hydrolase activity acting on glycosyl bonds (14), catalytic activity (76), peptidase activity (14), hydrolase activity acting on carbon–nitrogen (but not peptide) bonds (6), serine hydrolase activity (5), serine-type peptidase activity (5), metallopeptidase activity (5), structural constituent of ribosome (6), and flavin–adenine–dinucleotide binding (6) for MF (Figure 5, Appendix A). These DEPs may be involved in the 1% carbon dioxide concentration.

Carbohydrate, organic substance, peptide, and other metabolic processes were active in the 1% carbon dioxide concentration group (HCO), which indicates that the cellular metabolic process is active in high-carbon-dioxide-concentration conditions. From the molecular function aspect, hydrolase activity, catalytic activity, peptidase activity, the structural constituent of ribosome, and flavin–adenine–dinucleotide binding also confirm that material transformation and metabolism occur frequently in *P. ostreatus*.

### 3.4. Weighted Gene Co-Expression Network Analysis for DEPs

In total, 1333 proteins were classified into seven different modules by weighted gene co-expression network analysis (WGCNA) (602 proteins in the largest turquoise module, 374 in blue, 97 in yellow, 97 in brown, 68 in grey, 55 in green, and 40 in red). Eigengenes of each module in each treatment show different expression levels of genes in each module. Proteins in the MEblue module show relatively high expression levels in HCO compared with LCO, whereas proteins in the MEturquoise module show relatively high expression levels in LCO compared with HCO (Figure 6 and Figure 7, Appendix A).

The eigengene adjacency heatmap shows that MEred and MEgreen clustered with MEturquoise and MEyellow. MEblue clustered with MEbrown. There were 20 hub genes in six modules, wherein 43 proteins were uncharacterized proteins (blue, brown, green, red, turquoise, yellow). Many modules showed more or less intersection in gene function, especially between MEbrown and MEblue. Ribosomal protein was shared in MEbrown, MEblue, and MEyellow modules. Alkaline phosphatase was shared in MEbrown and MEblue modules. Autophagy-related protein was shared in MEblue and MEred modules. Extracellular metalloproteinase was shared in MEblue and MEbrown modules. Glycoside hydrolase was shared in MEgreen, MEbrown, and MEred modules. Glycosyltransferase was shared in MEgreen and MEturquoise modules (Figure 8 and Figure 9, Appendix A).

In the MEblue module, hub genes function as extracellular metalloproteinase, alkaline phosphatase, lipase_3 domain-containing protein, ribosomal protein, autophagy-related protein, hydrolase-1 domain-containing protein, chitinase, cellulase, and Oxidored_FMN domain containing protein, etc., which are related to HCO high expression compared with LCO. In the MEbrown module, hub genes function as ribosomal protein, glycoside hydrolase, tubulin, alkaline phosphatase, extracellular metalloproteinase, and RNA-binding domain-containing protein, etc. In the MEgreen module, hub genes function as glycoside hydrolase, glycosyltransferase, NADH-cytochrome b5 reductase, ribonuclease, Iso_dh domain-containing protein, dipeptidyl-peptidase, aminomethyltransferase, AMP-binding domain-containing protein, pyruvate kinase, tyrosine -tRNA ligase, chitin synthase, CN hydrolase, NAD(P)-bd_dom domain-containing protein, and FMN hydroxy acid dehydrogenase, etc. 

## 4. Discussion

CO_2_ is an end product of cellular respiration [14]. CO_2_ is a critical cellular signaling molecule in all organisms. In most basic aspects of life, the transport of CO_2_ through membranes has fundamental roles. High CO_2_ concentrations are sensed by cells independent of O_2_ of pH via specific signaling pathways, causing distinct effects (phenotypes) [15]. The CO_2_-induced enhancement of plant growth indicates that rising atmospheric CO_2_ has contributed to the shrubland expansions of the past 200 years [10]. CO_2_ plays a key role in respiration in mammals, microbial photosynthesis in plants and algae, and chemoreception in insects. During sexual reproduction, CO_2_ inhibits cell–cell fusion but not filamentation [18]. Elevated CO_2_ directly or indirectly influences plant–biotic interactions. For instance, elevated CO_2_ alters reactive oxygen signaling, phytohormone secondary metabolism, and defense-associated development. Elevated CO_2_ also directly or indirectly influences herbivory- or pathogenesis-related traits in pest and pathogen populations and alters predator–prey interactions by interfering with chemical communications and indirect defenses in pests [20]. In a previous study, carbon dioxide was shown to influence the initiation of the fruiting body development of mushrooms such as *Schizophyllum commune*, *F. filiformis*, and *Agaricus bisporus* [29,30]. However, research on the responses of *P. ostreatus* under different carbon dioxide concentrations is limited. The inhibition of pileus expansion by high CO_2_ levels was especially conspicuous in *P. ostreatus*, causing a trumpet-shaped deformation of the pileus, occasional swelling of the stipe accompanied by sponge-like tissue, and, occasionally, malformed small pileus surfaces.

The expansion of the pileus, the basidiospore-forming area, is inhibited at high CO_2_ concentrations. As we can see from the results, the expression of the sexual differentiation process protein Isp4 (A0A067N4P4) was significantly down-regulated in the HCO group. Sexual differentiation process proteins have been studied in *Schizosaccharomyces pombe* and some other fungi [31,32], and isp4 has been found to be up-regulated by nitrogen-starvation-induced meiosis. A0A067N4P4, which might be involved in meiosis and reproduction in *P. ostreatus*, has 53.0% amino-acid sequence identity with Isp4 in *S. pombe* (NP_595653). The unmatured pileus at high CO_2_ concentrations also inhibits the sexual production process. In the natural environment, a high CO_2_ concentration indicates poor ventilation, which is not good for basidiospore dispersal. Therefore, the mushroom might control the sexual process by controlling the fruiting body morphology.

GO enrichment indicated that enzymes related to hydrolase activity [GO:0016810] were significantly enriched (with the lowest *p*-value) at a high CO_2_ concentration. Several amidohydrolases were significantly highly expressed in the HCO group, and no amidohydrolase was significantly down-regulated in the LCO group. Alfonso et al. reported that amidohydrolase might be related to autolysis in penicillin, and Donovan et al. reported that amidohydrolase might be related to cell wall lysis [33,34]. Therefore, the high expression of amidohydrolase in the HCO group indicates that cell wall biosynthesis might be influenced by a high CO_2_ concentration. Indeed, some cell-wall-synthesis-related proteins were also significantly highly expressed at a high CO_2_ concentration (A0A067NEG7), such as chitooligosaccharide oxidase (A0A067NL12), chitin deacetylase, and chitinase-3-like protein 2 (A0A067NHR3). The results indicate that the morphological change in *P. ostreatus* at a high CO_2_ concentration was regulated by the differential expression of cell wall biosynthesis.

Kinase-mediated protein phosphorylations play pivotal roles in the regulation of cellular processes. Physiological CO_2_ concentrations induce filamentation by the direct stimulation of cyclase activity. Filamentation is mediated by second messengers, such as cyclic adenosine 3′,5′-monophosphate (cAMP) synthesized by adenylyl cyclase. The link between cAMP signaling and CO_2_ sensing is conserved in fungi, and the cAMP signaling pathway in turn controls the growth, differentiation, and virulence factors of fungi [35,36]. Different kinases may be involved in the subsequent regulation of cellular processes. In high CO_2_ concentrations, the expression of many kinases was significantly down-regulated, such as A0A067NLY8, A0A067NNC2, A0A067NJ97, A0A067NHQ7, A0A067N5R0, A0A067NR52, A0A067NIN8, A0A067PB97, A0A067N9N0, A0A067NAM8, and A0A067NE46. However, no predicted kinase was highly expressed under high CO_2_ concentrations. In this study, the sample for proteomic analysis is the pileus part. The small pileus combined with the proteomic analysis results suggests that the metabolic activity is lower in high CO_2_ conditions. High CO_2_ concentrations might inhibit the expression of kinases, which further inhibit the metabolism and development of the fruiting body pileus.

Seven elongation factors were identified in proteomic analysis (Appendix A): the expression of five elongation factors was significantly down-regulated in high CO_2_ concentrations, the expression of one elongation factor was insignificantly down-regulated in high CO_2_ concentrations, and one was only expressed in low CO_2_ concentration. Translation elongation factor complexes play a central role in protein synthesis, delivering aminoacyl-tRNAs to the elongating ribosome, and the expression of elongation factor is indispensable in eukaryotes [37,38,39]. The down-regulation of elongation factors at high CO_2_ concentrations might inhibit the protein synthesis, which is inconsistent with the down-regulation of kinases. Therefore, the elongation factors might be involved in CO_2_-mediated morphological change in *P. ostreatus*.

## 5. Conclusions

This research investigates the proteome analysis of commercially relevant edible fungi *P. ostreatus* under different carbon dioxide concentrations. The stipes were shorter and thinner in the HCO group compared with the LCO group. A larger pileus diameter was observed in the LCO group compared with the HCO group. The differentially expressed proteins under higher and lower CO_2_ concentrations were presented and discussed in the present study. Based on our results, the expression of kinases and elongation factors are regulated under different CO_2_ concentrations, and, as a result, the expressions of proteins related to cell wall synthesis and sexual differentiation process proteins are significantly changed. Proteins in these processes can serve as a target for selective molecular breeding. Further studies are still needed to analyze the function of other proteins, such as proteins related to cAMP signal processing or bicarbonate metabolism, in CO_2_ response in *P. ostreatus*.

## Figures and Tables

**Figure 1 jof-08-00652-f001:**
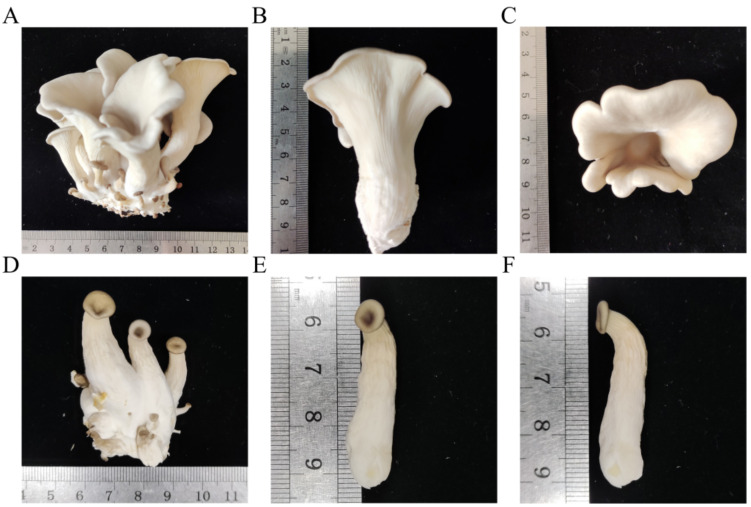
Morphological comparison of fruiting bodies of *P. ostreatus* PH11 under different carbon dioxide concentrations. (**A**–**C**): The fruiting body of *P. ostreatus* cultured under atmospheric conditions (LCO). (**D**–**F**): The fruiting body of *P. ostreatus* cultured under 1% carbon dioxide concentration (HCO). (**A**,**D**) Morphology, pileus size of *P. ostreatus* under low (**A**) or high (**D**) CO_2_ concentrations. (**B**,**E**) Stipe lengths of *P. ostreatus* under low (**B**) or high (**E**) CO_2_ concentrations, and (**C**,**F**) pileus diameters of *P. ostreatus* under low (**C**) or high (**F**) CO_2_ concentrations.

**Figure 2 jof-08-00652-f002:**
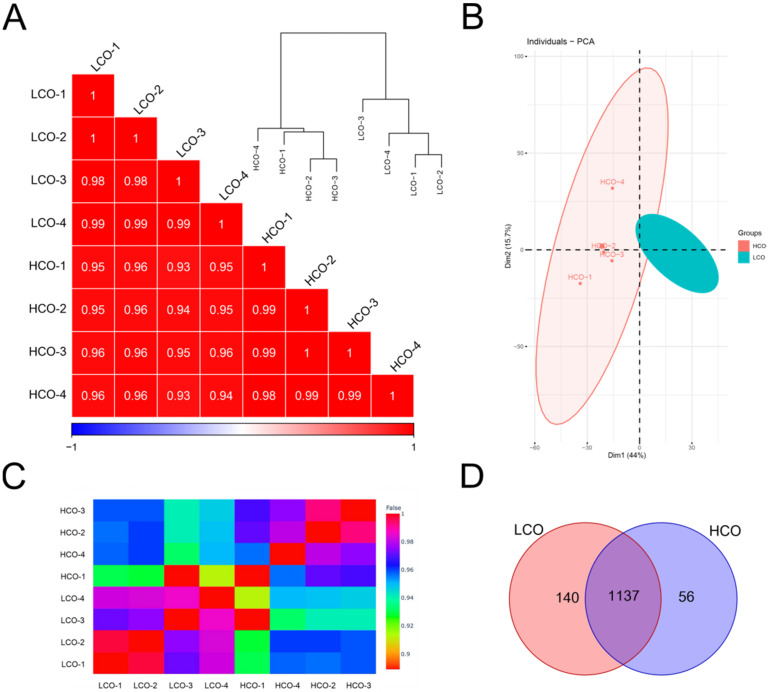
PCC analysis and PC analysis representation of the proteomic dataset of two groups of samples. (**A**) Pearson correlation coefficient analysis for pair-wise comparisons of proteome data. (**B**) Principal component analysis of proteome data from two groups of samples. The HCO group and LCO group are located separately. (**C**) Heatmap plot for pair-wise comparisons of proteome data. (**D**) Venn diagram of the two groups of samples.

**Figure 3 jof-08-00652-f003:**
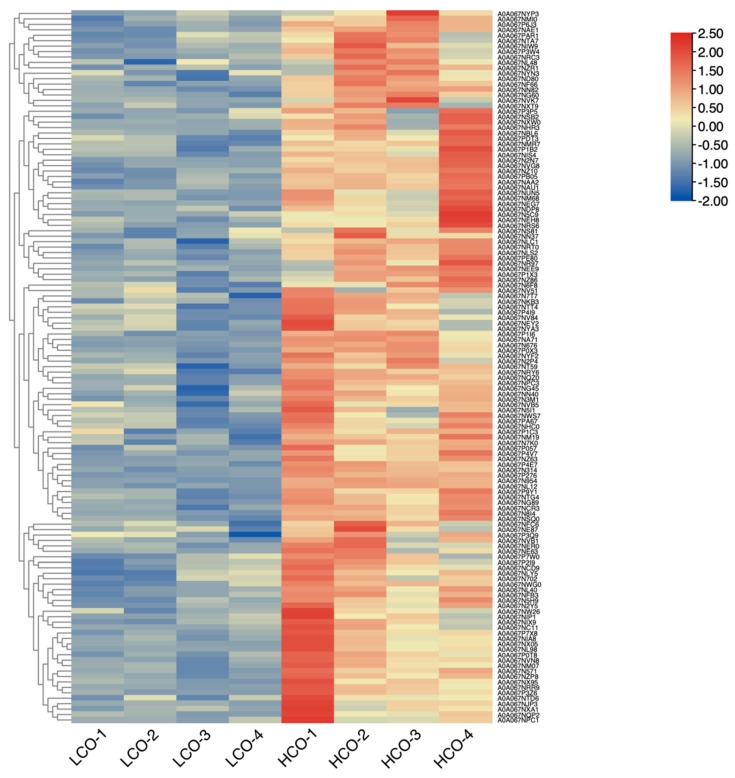
Heatmap of 131 highly expressed proteins in HCO group compared with LCO group in *P. ostreatus* PH11.

**Figure 4 jof-08-00652-f004:**
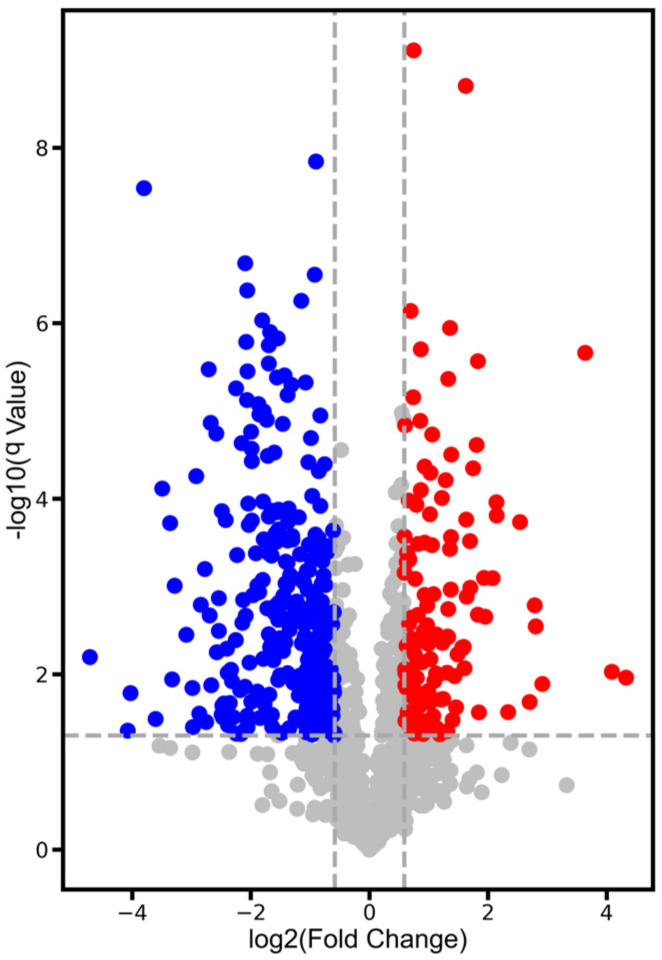
Volcano plot of differentially expressed proteins in HCO group compared with LCO group in *P. ostreatus*. Red dots indicate up-regulated proteins in LCO group. Blue dots indicate up-regulated proteins in HCO group. Grey dots indicate no significantly differentially expressed proteins. The threshold for differentially expressed proteins is a fold-change more than 1.5-fold and FDR < 0.05.

**Figure 5 jof-08-00652-f005:**
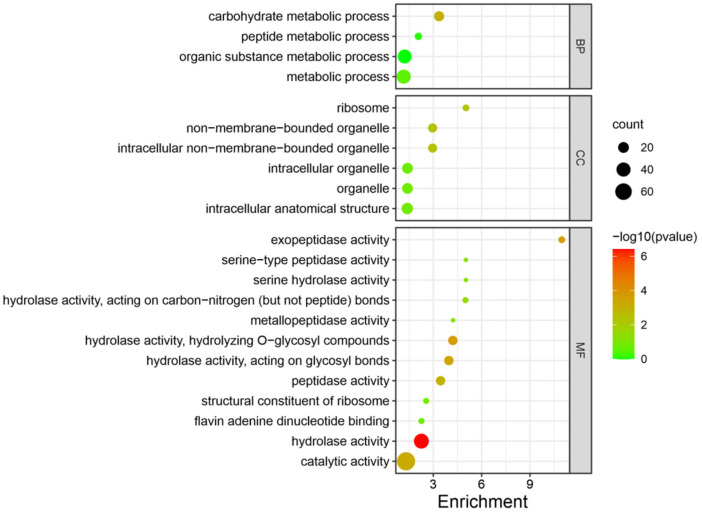
Enriched GO terms of 131 highly expressed proteins in HCO group compared with LCO group in *P. ostreatus* PH11. Bubble size indicates protein counts. Bubble color indicates *p*-value. BP represents biological process, CC represents cellular component, MF represents molecular function.

**Figure 6 jof-08-00652-f006:**
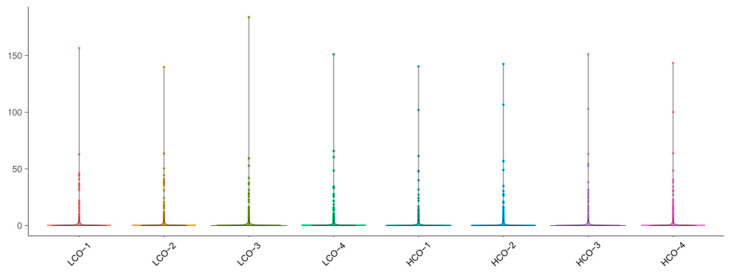
Sample distribution pattern of eight samples.

**Figure 7 jof-08-00652-f007:**
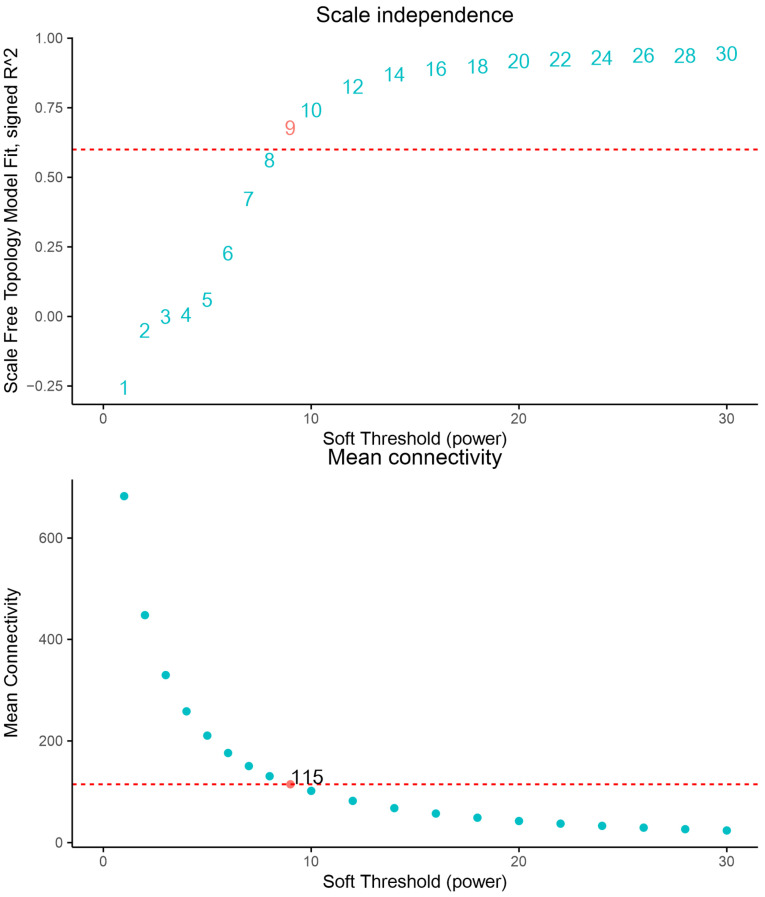
Softpower of weighted gene co-expression network analysis. Nine were finally chosen as the power for WGCNA analysis.

**Figure 8 jof-08-00652-f008:**
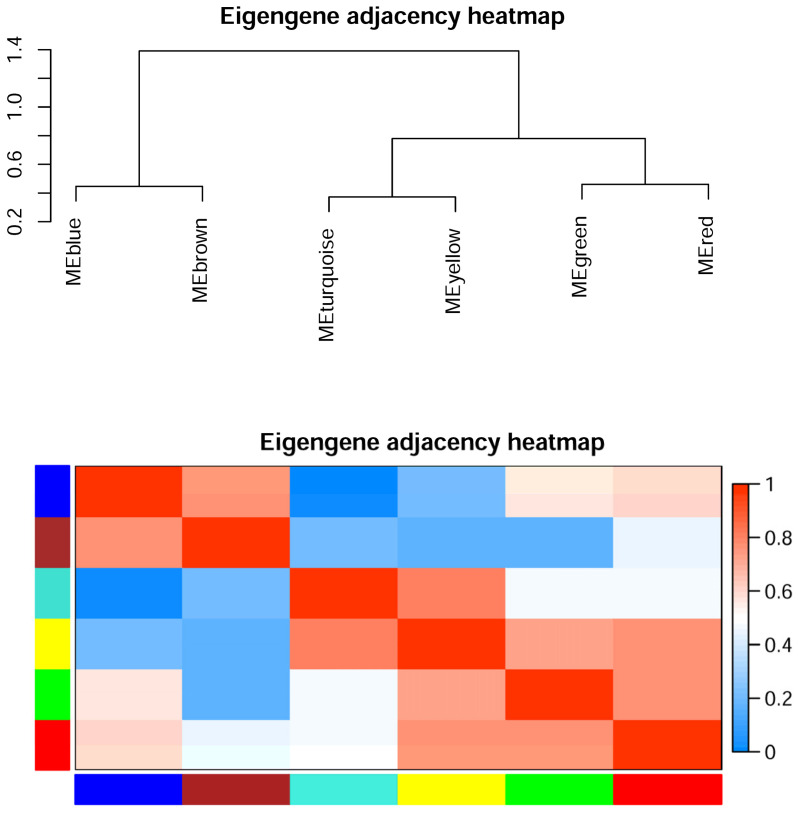
Module correlation plot of weighted gene co-expression network analysis. Eigengene is the first principal component gene of the specific module, representing the overall gene expression level of the module.

**Figure 9 jof-08-00652-f009:**
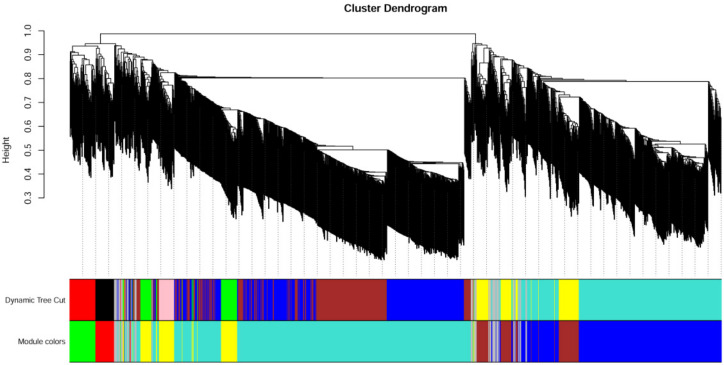
Module correlation plot of weighted gene co-expression network analysis. Genes are clustered by dissimilarity between genes, and the gene tree is divided into different modules.

## Data Availability

The raw data for the proteomic analysis reported in this paper have been deposited in the OMIX, China National Center for Bioinformation/Beijing Institute of Genomics, Chinese Academy of Sciences (https://ngdc.cncb.ac.cn/, accessed on 26 April 2022, omix: accession no. OMIX001097).

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
