# Peer review of "Responses of the Mushroom Pleurotus ostreatus under Different CO2 Concentration by Comparative Proteomic Analyses"

_jof, 2022, doi:10.3390/jof8070652_

Round 1
Reviewer 1 Report
at the present form.
Kindly use the organism's full name at first occurrence (Pleurotus ostreatus) of the MS; later in all occurrences use the short (P. ostreatus) name (in italics) only.
Eg: cultivated mushroom varieties of F. velutipes, P. ostreatus, 102
growth of hypha of Pleurotus ostreatus. Afterwards, 120
Pleurotus ostreatus Samples 133
Vegetable growth of P. ostreatus my- 138
After adding 15 mg DTT, the sample…. Kindly define the abbreviation of the chemical at their first occurrences or under a separate subheading like “chemicals”.
Pearson correlation coefficient (PCC) analysis, and principal component 24
(PC) analysis were performed to reveal the correlation among samples. The differentially expressed proteins (DEPs) were organized. Gene Ontology (GO)…kindly remove the abbreviation in abstract sections unless it is repeated.
CO2 chamber with 90%–95% humidity,…. Kindly provide the details of CO2 chamber (like the company name and model).
grow under different CO2 concentration for 72 h, and the fruiting body were col- 144…..Please specify the CO2 concentration.
The formula of solid media was as follows: 55% cottonseed hull, 30% sawdust, 10% 136--- Please specify the sterilization of media?
- The control group bags were incubated in atmosphere under the same condition. …. Please specify the same condition and rewrite the sentence.
was incubated at 37 â—¦C for 1 h [30]. .....Check the degree symbol.
Label-Free LC-MS/MS Quantitative….. Please specify the temperature-programmed and need to add more details.
length (3.9cm, 9.5cm), much thinner in the HCO 198
group as compared with the LCO group…. Define HCO and LCO at their first occurrences.
CO2 conditions (Figure 1AD) change it like (Figure 1A-1D) in all occurrences.
much thinner (3.9cm) in the HCO group as compared with the LCO group (9.5 cm), (Figure 1B and 1E)…. Change this line like this.
Figure 1. Morphological comparison of fruiting bodies of P. ostreatus PH11 under different carbon dioxide concentrations. (A, B, and C): The fruiting body of P. ostreatus cultured under atmospheric conditions (LCO). (D, E, and F):The fruiting body of P. ostreatus cultured under 1% carbon dioxide concentration (HCO). (A and D) Morphology, pileus size, (B and E) stipe lengths, and (C and F) Pileus diameters of P. ostreatus. …………Change like this. Change all figure legends for better clarity.
expression profile of the LCO group and HCO group diplay a….display a… English spell and grammar checks are required for the entire manuscript.
Line no. 218….if all mushrooms are the same biological replicates, then why HCO-4 and HCO-1 vary significantly? Justify it.
Many DEPs are domain-containing proteins, which seem to be proteins with different biological functions. (Such as A0A067NG89, A0A067NZR1). Some are hydrolase. (Such as A0A067NG45, A0A067NTT4). Some are histone. (Such as A0A067NL48, A0A067P7W0). 247 Some are chitinase. (Such as A0A067N5C9, A0A067NX05). Some are ribosomal protein. 248 (Such as A0A067NG60, A0A067N2P4)……rewrite this line for better clearness.
organic substance metabolic process (39), peptide metabolic process (8)…… kindly check for space between the numbers and alphabets in all places …
The visibility of figure 6 is very poor. So kindly provide higher quality image.
The discussion section is weak, thus kindly improve the discussion section. The authors requested to compare and/or justify their findings with the literature.
The conclusion section must be improved. In conclusion section kindly highlights all findings in relation to industrial aspects.
Author Response
1) Kindly use the organism's full name at first occurrence (Pleurotus ostreatus) of the MS; later in all occurrences use the short (P. ostreatus) name (in italics) only. Eg: cultivated mushroom varieties of F. velutipes, P. ostreatus, 102 growth of hypha of Pleurotus ostreatus. Afterwards, 120 Pleurotus ostreatus Samples 133 Vegetable growth of P. ostreatus my- 138
Response:
We appreciate the reviewer’s comment. The use of full name or abbreviation of organisms need to be modified to make the language concise and brief. Avoidance of repetition and verbose is indispensable in writing papers. We have already used the organism's full name at first occurrence (Pleurotus ostreatus) of the MS; later in all occurrences use the short (P. ostreatus) name (in italics) only in the whole paper.
2) After adding 15 mg DTT, the sample…. Kindly define the abbreviation of the chemical at their first occurrences or under a separate subheading like “chemicals”.
Response:
We are really sorry for the confusion of these undefined abbreviations different chemicals. We have already defined the abbreviation of the chemical at the first occurrence. The sentence is modified as follows: After adding 15 mg dithiothreitol (DTT), the sample was incubated at 37â—¦C for 1 h [30].
100 mg frozen sample was taken into the centrifuge tubes and then 1 mL UT buffer (8 M urea, 0.1 M Tris-HCl pH 8.5) containing Thermo HALT protease and phosphatase inhibitor cocktail was added.
‘All gradient started at 5% (v/v) acetonitrile (ACN) (0.1% formic acid) and went up to 32% (v/v) ACN (0.1% formic acid).’ In ‘2.3. Label-Free LC-MS/MS Quantitative Proteomics Analysis’ part.
3) Pearson correlation coefficient (PCC) analysis, and principal component 24 (PC) analysis were performed to reveal the correlation among samples. The differentially expressed proteins (DEPs) were organized. Gene Ontology (GO)…kindly remove the abbreviation in abstract sections unless it is repeated.
Response:
We appreciate the reviewer’s comment. We really agree that removing the abbreviation in abstract sections is still needed to make the language simplification. We have already removed the abbreviation in abstract sections unless it is repeated.
4) CO2 chamber with 90%–95% humidity,…. Kindly provide the details of CO2 chamber (like the company name and model).
Response:
We really appreciate the reviewer’s critical and thoughtful comments. Details are necessary for CO2 chamber. Fruiting production at high CO2 concentration was carried out at ZCLY-180ES CO2 chamber (Zhichu, Shanghai) with 90%-95% humidity. The humidity and light was controlled by wireless humidifier and light.
5) grow under different CO2 concentration for 72 h, and the fruiting body were col- 144…..Please specify the CO2 concentration.
Response:
We really appreciate the reviewer’s critical and thoughtful comments. The reviewer is an expert in the edible mushroom area. It is necessary to specify the CO2 concentration to make the methods clear and in detailedness: We have already modified in under different CO2 concentration (1%,0%)
6) The formula of solid media was as follows: 55% cottonseed hull, 30% sawdust, 10% 136--- Please specify the sterilization of media?
Response:
We really appreciate the reviewer’s critical and thoughtful comments. Details are necessary for the sterilization of media. For fruiting body production, strain PH11 was inoculated into solid media in polypropylene bags (5 cm * 30 cm, 50 μm thinkness). The formula of solid media was as follows: 55% cottonseed hull, 30% sawdust, 10% bran, 3% gypsum, 0.5% potassium dihydrogen phosphate, 0.5% urea, 1% glucose, and the ratio of material to water was 1:1.5, pH was natural. The cultivation bags with solid media were sterilized at 121 °C for 2 hours, incubated at root temperature for 72 hours, and sterilized at 120 °C for 2 hours for the second time.
7) The control group bags were incubated in atmosphere under the same condition. …. Please specify the same condition and rewrite the sentence. was incubated at 37 â—¦C for 1 h [30]. .....Check the degree symbol.
Response:
We really appreciate the reviewer’s critical and thoughtful comments on methods. We have already rewritten the sentence. The original sentence: The control group bags were incubated in atmosphere under the same condition. The modified sentence: The control group bags were incubated in atmosphere with 90%–95% humidity, 12 h light/ 12 h dark, and at 20°C.
We are really sorry for the wrong degree symbol. We have already modified: After adding 15 mg DTT, the sample was incubated at 37â—¦C for 1 h.
8) Label-Free LC-MS/MS Quantitative….. Please specify the temperature-programmed and need to add more details.
Response:
We really appreciate the reviewer’s critical and thoughtful comments. Details are necessary for temperature-programmed. This analysis was performed with desalted peptides, which were reconstituted in 10 μL 0.1% formic acid, which was carried out using a Nano-LC system coupled with Orbitrap FusionTM TribridTM (Thermo Fisher Scientific). The peptide sample (1 μL) were injected onto a Acclaim PepMap 100 nano trap column (75μm×2 cm, nanoViper 2PK C18, 3 μm, 100 Å) and the separated on a Acclaim PepMapTM 100 analytical column (75μm×15 cm, nanoViper C18, 3 μm, 100 Å), and eluted in a 60 min gradient of 5-40 % ACN in 0.1% formic acid at 300 nL/min, followed by 5 min ramping to 80% ACN/0.1%formic acid and hold at 80% ACN/0.1%formic acid for 10 min. The Orbitrap Fusion is operated in positive ion mode with spray voltage set at 2.2 KV and source temperature at 275℃. The MS instrument was operated in data-dependent acquisition mode (DDA), with full MS scans over a mass range of m/z 350–1500 with detection in the Orbitrap (120 K resolution) and with auto gain control (AGC) set to 100,000 or a maximum ion injection time of 50 ms. For survey scan, ten of the most abundance precursor ions with a charge state of 2+ to 6+ were selected for higher energy collisional dissociation (HCD) fragment analysis. The dynamic exclusion parameter was set at 60 s. For each group, four biological repeats were performed.
9) length (3.9cm, 9.5cm), much thinner in the HCO 198 group as compared with the LCO group…. Define HCO and LCO at their first occurrences.
Response:
We are really sorry for the confusion of these undefined abbreviations at their first occurrences. We have modified as follows: however, the stipes were much shorter in length (3.9cm, 9.5cm), much thinner in the 1% carbon dioxide concentration (HCO) group as compared with the atmospheric condition (LCO) group.
10) CO2 conditions (Figure 1AD) change it like (Figure 1A-1D) in all occurrences. much thinner (3.9cm) in the HCO group as compared with the LCO group (9.5 cm), (Figure 1B and 1E)…. Change this line like this.
Response:
We really appreciate the reviewer’s critical and thoughtful comments. The reviewer is an expert in the edible mushroom area. It is necessary to label the figure clearly. We have already modified from ‘(Figure 1AD)’ to ‘(Figure 1A-1D)’, from ‘(Figure 1BE)’ to ‘(Figure 1B and 1E)’, from ‘(Figure 1CF)’ to ‘(Figure 1C and 1F)’ in ‘3.1. Phenotype of P. ostreatus under Different CO2 Concentrations’ part.
11) Figure 1. Morphological comparison of fruiting bodies of P. ostreatus PH11 under different carbon dioxide concentrations. (A, B, and C): The fruiting body of P. ostreatus cultured under atmospheric conditions (LCO). (D, E, and F): The fruiting body of P. ostreatus cultured under 1% carbon dioxide concentration (HCO). (A and D) Morphology, pileus size, (B and E) stipe lengths, and (C and F) Pileus diameters of P. ostreatus. …………Change like this. Change all figure legends for better clarity.
Response:
We really appreciate the reviewer’s critical and thoughtful comments on the format of figure legends. We have already changed all figure legends for better clarity. Figure 1. Morphological comparison of fruiting bodies of P. ostreatus PH11 under different carbon dioxide concentrations. (A,B, and C): The fruiting body of P. ostreatus cultured under atmospheric conditions (LCO). (D,E, and F): The fruiting body of P. ostreatus cultured under 1% Carbon dioxide concentration (HCO). (A and D) Morphology, pileus size of P. ostreatus under low (A) or high (D) CO2 concentrations. (B and E) Stipe lengths of P. ostreatus under low(B) or high(E) CO2 concentrations, and (C and F) Pileus diameters of P. ostreatus under low(C) or high(F) CO2 concentrations.
12) expression profile of the LCO group and HCO group diplay a….display a… English spell and grammar checks are required for the entire manuscript.
Response:
We really appreciate the reviewer’s critical and thoughtful comments on English spelling and grammar. English spell and grammar checks are necessary for the entire manuscript. We have already checked English spelling and grammar of the whole paper. The protein expression profile of the LCO group and HCO group diplays a higher correlation (>0.93) with each other.
13) Line no. 218….if all mushrooms are the same biological replicates, then why HCO-4 and HCO-1 vary significantly? Justify it.
Response:
We really appreciate the reviewer’s critical and thoughtful comments on biological replicates. All mushrooms are the same biological replicates. But HCO-4 and HCO-1 vary significantly due to sampling and sequencing deviation affected by the environment. Next time we will improve our sampling and sequencing scheme to make all the biological replicates vary a little.
14) Many DEPs are domain-containing proteins, which seem to be proteins with different biological functions. (Such as A0A067NG89, A0A067NZR1). Some are hydrolase. (Such as A0A067NG45, A0A067NTT4). Some are histone. (Such as A0A067NL48, A0A067P7W0). 247 Some are chitinase. (Such as A0A067N5C9, A0A067NX05). Some are ribosomal protein. 248 (Such as A0A067NG60, A0A067N2P4)……rewrite this line for better clearness. organic substance metabolic process (39), peptide metabolic process (8)…… kindly check for space between the numbers and alphabets in all places …
Response:
We really appreciate the reviewer’s critical and thoughtful comments on format. We have already checked for space between the numbers and alphabets in all places in ‘3.2. Proteomic Analysis of P. ostreatus Grown under Different CO2 Conditions’ part. The modified part: A wide variety of DEPs are annotated with different biological functions. For instances, some DEPs are hydrolases (Such as A0A067NG45, A0A067NTT4), which catalyze the hydrolysis of a chemical bond. Ribosomal proteins, such as A0A067NG60, A0A067N2P4, play roles in the formation and functioning of the ribosome. Chitinases, including A0A067N5C9, A0A067NX05, play significant roles in stipe cell wall extension in mushroom. Some DEPs are histones (Including A0A067NL48, A0A067P7W0), which provide structural support for a chromosome.
15) The visibility of figure 6 is very poor. So kindly provide higher quality image.
Response:
Thank you very much for the detailed suggestion. We are really sorry for the confusion and poor visibility of figure 6. We have already provided a higher quality image of figure 6 as follows.
16) The discussion section is weak, thus kindly improve the discussion section. The authors requested to compare and/or justify their findings with the literature.
Response:
We really appreciate the reviewer’s critical and thoughtful comments on discussion section. We have already compared our findings with the literature. We have already improved the discussion section in the article. We also discussed higly expression autophagy-related proteins in high carbon dioxide concentration group: Autophagy: Autophagy-related protein 12 (Ubiquitin-like protein ATG12) (A0A067N571): autophagosome assembly [GO:0000045], cytoplasm [GO:0005737]; O-methyltransferase activity [GO:0008171]. Ubiquitin-like modifier-activating enzyme ATG7 (Autophagy-related protein 7) (A0A067NLY5): autophagy of mitochondrion [GO:0000422]; C-terminal protein lipidation [GO:0006501]; late nucleophagy [GO:0044805]; macroautophagy [GO:0016236]; piecemeal. cytosol [GO:0005829]; extrinsic component of phagophore assembly site membrane [GO:0097632]. Atg12 activating enzyme activity [GO:0019778]; Atg8 activating enzyme activity [GO:0019779]; identical protein binding [GO:0042802]. Autophagy-related structures in P. ostreatus and possible model for autophagosome formation in filamentous fungi were provided in previous research [34]. Autophagy is associated with a wide variety of fungal morphogenetic processes.
17) The conclusion section must be improved. In conclusion section kindly highlights all findings in relation to industrial aspects.
Response:
We really appreciate the reviewer’s critical and thoughtful comments on conclusion section. We have already highlighted all findings in relation to industrial aspects: This research investigates s the proteome analysis of commercially relevant edible fungi P. ostreatus under different carbon dioxide concentration. These DEPs under high CO2 level might play vital roles in metabolic processes and activities. The DEPs and GO terms can add value to the understanding of mechanisms concerning responses of filamentous fungi under different CO2 concentration during commercial prodution. The stipes were much shorter, much thinner in the HCO group compared with LCO group. A larger pileus diameter was observed in LCO group compared with HCO group. Customers prefer P. ostreatus with larger pileus (LCO) and shorter stipe (LCO). Therefore, in industrial production, the CO2 concentration can be controlled between 0% and 1% in order to produce high quality P. ostreatus. Different metabolic and catalytic activities in HCO group contribute to understanding the more active biological processes in industrial production.
Reviewer 2 Report
The manuscript describes the effect of carbon dioxide on the fruiting of mushrooms using proeomic analysis. It carries some interesting findings and may contribute understabding of the fruiting body formation of mushrooms. However, it needs a significant modification before its' final decision. The introduction part contains many redundent descriptions on the effect of CO2. Result section from the subsection 3.4 does not fully explain their findings in terms of protein functions related to the fruiting. It simply categorizes the proteins in different groups. Each figure is not fully explained ehther in the main text or in the figure legend.
There are many errors in the text writing. Some of them are as follows:
L20: remove “The” in front of “Pleurotus ostreatus”
L21, L29, L30, L31, L37: P. ostreatus
L21: is few.
L35: provide full description of WGCNA
L53: studies, affects
L63: Lu et al.
L66: pileus, which would be
L67: put a space between “process” and “[13-17]”. L75, L85 too.
L102: Flammulina velutipes, Pleurotus ostreatus
L103: italicize “Pholiota nameko, Lentinula edodes.
L111: P. eryngii
L120, L123 and below: P. ostreatus
L122: The lower limit of the stimulating effect is 22% and the highest limit is 37.5%
L135: the strain
L137: , and 1% glucose. The ratio of substrates to water was 1:1.5. pH was neutral.
L138: Vegetative growth, italicize “ P. ostreatus”
L140: , and light in high humidity.
L143: The fruiting bodies were grown
L148: protocol. 100 mg frozen
L155, L164, L170: 37oC, -80oC
L159, L165: 10,000 xg
L161: what is UA?
L166: 50 μL of 50 mM
L167: Salt in the pooled
L179: provide full description for ACN
L196: high concentration (1%) of
l245-L249: rewrite this part.
L273-L274: why do you include this paragraph? it is unecessary. Otherwise, describe it with citation.
L280: Capitalize each word.
L281: weighted gene co-expression network analysis (WGCNA)
L282,L283: classified
L282-L286: rewrite this part.
Author Response
1) The manuscript describes the effect of carbon dioxide on the fruiting of mushrooms using proeomic analysis. It carries some interesting findings and may contribute understabding of the fruiting body formation of mushrooms. However, it needs a significant modification before its' final decision. The introduction part contains many redundent descriptions on the effect of CO2. Result section from the subsection 3.4 does not fully explain their findings in terms of protein functions related to the fruiting. It simply categorizes the proteins in different groups. Each figure is not fully explained ehther in the main text or in the figure legend. There are many errors in the text writing. Some of them are as follows: L20: remove “The” in front of “Pleurotus ostreatus”
Response:
We really appreciate the reviewer’s critical and thoughtful comments. In abstract part, ‘Pleurotus. ostreatus is a popular edible mushroom in East Asia markets.’
2) L21, L29, L30, L31, L37: P. ostreatus
Response:
We appreciate the reviewer’s comment. The use of full name or abbreviation of organisms need to be modified to make the language concise and brief. Avoidance of repetition and verbose is indispensable in writing papers. We have already used the organism's full name at first occurrence (Pleurotus ostreatus); later in all occurrences use the short (P. ostreatus) name (in italics) only in the whole paper.
3) L21: is few.
Response:
We really appreciate the reviewer’s critical and thoughtful comments. This part has been deleted from the manuscript.
4) L35: provide full description of WGCNA
Response:
We really appreciate the reviewer’s critical and thoughtful comments. We have already provided full description of WGCNA in abstract part as follows: 1333 proteins were classified into seven different modules by WGCNA, wherein 602 proteins classify into turquoise module. In MEblue module , hub genes function as extracellular metalloproteinase, alkaline phosphatase, lipase, ribosomal protein, autophagy-related protein, hydrolase, chitinase, cellulase, Oxidored_FMN, etc, which are related to HCO highly expression compared with LCO.
5) L53: studies, affects
Response:
We really appreciate the reviewer’s critical and thoughtful comments. Environmental factors for fruiting body induction (light, temperature, and nutrients as three examples), development (light, gravity, and carbon dioxide concentration as three examples), and maturation or senescence (during/after sporulation and after artificial harvesting) of mushroom-forming basidiomycetes are researched in the previous studies, wherein carbon dioxide affects development of mushroom
6) L63: Lu et al.
Response:
According to the reviewer’s suggestion, we have already change ‘et al’ into ‘et al.’: Lu et al. unraveled a new regulatory mechanism of CO2 signaling in fungi hyphal development by reducing Ume6 phosphorylation and degradation.
7) L66: pileus, which would be
Response:
According to the reviewer’s suggestion, we have already modified the sentence as follows: CO2 concentration could affect the development of fruiting bodies. What is more, the differentiation of pileus would be inhibited under high CO2 concentration.
8) L67: put a space between “process” and “[13-17]”. L75, L85 too.
Response:
We appreciate the reviewer’s comment on format. We have already put a space between the last word of a sentence and the reference number , such as “process” and “[13-17]”.
9) L102: Flammulina velutipes, Pleurotus ostreatus L103: italicize “Pholiota nameko, Lentinula edodes.
Response:
According to the reviewer’s suggestion, we have already modified the sentence as follows: ‘Sensitivity to carbon dioxide has been investigated in the commercially cultivated mushroom varieties of Flammulina velutipes, Pleurotus ostreatus, Pholiota nameko (microspora), Lentinula edodes; of these, wherein P. ostreatus is most sensitive to carbon dioxide.’ In ‘1.Introduction’ part.
10) L111: P. eryngii
Response:
We really appreciate the reviewer’s critical and thoughtful comments. This part has been deleted from the manuscript.
11) L120, L123 and below: P. ostreatus
Response:
We really appreciate the reviewer’s critical and thoughtful comments. This part has been deleted from the manuscript.
12) L122: The lower limit of the stimulating effect is 22% and the highest limit is 37.5%
Response:
We really appreciate the reviewer’s critical and thoughtful comments. This part has been deleted from the manuscript.
13) L135: the strain
Response:
According to the reviewer’s suggestion, we have already modified the sentence as follows: ‘For fruiting body production, the strain PH11 was inoculated into solid media in polypropylene bags.’ in ‘2.1. Culture Conditions and Acquisition of the P. ostreatus Samples’ part.
14) L137: , and 1% glucose. The ratio of substrates to water was 1:1.5. pH was neutral.
Response:
According to the reviewer’s suggestion, we have already modified the sentence as follows: ‘55% cottonseed hull, 30% sawdust, 10% bran, 3% gypsum, 0.5% potassium dihydrogen phosphate, 0.5% urea, and 1% glucose. The ratio of substrates to water was 1:1.5, pH was natural.’ in ‘2.1. Culture Conditions and Acquisition of the P. ostreatus Samples’ part.
15) L138: Vegetative growth, italicize “ P. ostreatus”
Response:
According to the reviewer’s suggestion, we have already modified the sentence as follows: ‘Vegetative growth of P. ostreatus mycelia was performed at 25°C with a humidity of 70% in darkness.’ in ‘2.1. Culture Conditions and Acquisition of the P. ostreatus Samples’ part.
16) L140: , and light in high humidity.
Response:
According to the reviewer’s suggestion, we have already modified the sentence as follows: ‘After 40 days cultivation, the primordium was stimulated by water injection, low temperature, and light in high humidity environment.’ in ‘2.1. Culture Conditions and Acquisition of the P. ostreatus Samples’ part.
17) L143: The fruiting bodies were grown
Response:
According to the reviewer’s suggestion, we have already modified the sentence as follows: ‘The fruiting bodies were grown under different CO2 concentration(1%,0%) for 72 h.’ in ‘2.1. Culture Conditions and Acquisition of the P. ostreatus Samples’ part.
18) L148: protocol. 100 mg frozen
Response:
According to the reviewer’s suggestion, we have already changed the comma into period between ‘protocol’ and ‘100 mg frozen’. ‘Total proteins were extracted from the frozen P. ostreatus samples according to the following protocol. 100 mg frozen sample was taken into the centrifuge tubes and then 1 mL UT buffer (8 M urea, 0.1 M Tris-HCl pH 8.5) containing HaltTM Protease inhibitor Cocktail was added.’ in ‘2.2 Protein Extraction and Peptide Digestion’ part.
19) L155, L164, L170: 37oC, -80oC
Response:
We appreciate the reviewer’s comment on format and punctuation, we have already modified the sentence as follows: ‘at 37â—¦C for 1 h’, ‘with gentle shaking at 37â—¦C for 12 h’, ‘stored at −80â—¦C’. in ‘2.2 Protein Extraction and Peptide Digestion’ part.
20) L159, L165: 10,000 xg
Response:
We appreciate the reviewer’s comment on format, we have already modified the sentence as follows: ‘remove the low molecular impurities by centrifuging at 10,000 ×g for 30 min.’. ‘peptides were collected by centrifugation at 10,000 ×g for 15 min.’ and ‘12,000 ×g for 10 min at 4℃’ in ‘2.2 Protein Extraction and Peptide Digestion’ part.
21) L161: what is UA?
Response:
We are really sorry for the confusion. UA means UA Buffer, which contains 8 m urea, 0.1 m Tris-HCl (pH 8.5) and 50 mm dithiothreitol.
22) L166: 50 μL of 50 mM
Response:
We appreciate the reviewer’s comment on grammar, we have already modified the sentence as follows: the residue peptides in the ultrafiltration tube were washed with 50 μL of 50 mM NH4HCO3 for one more time. We have already added ‘of ’ between ‘50 μL’ and ‘50 mM’ in ‘2.2 Protein Extraction and Peptide Digestion’ part.
23) L167: Salt in the pooled
Response:
We appreciate the reviewer’s comment on grammar, we have already modified the sentence as follows: ‘The salt in the pooled elutes was removed by using Merck Millipore ZipTip C18 resin (Darmstadt).’ We have already changed ‘of’ to ‘in’ in ‘2.2 Protein Extraction and Peptide Digestion’ part.
24) L179: provide full description for ACN
Response:
We are really sorry for the confusion. ACN is the abbreviation of acetonitrile. ‘All gradient started at 5% (v/v) acetonitrile (ACN) (0.1% formic acid) and went up to 32% (v/v) ACN (0.1% formic acid).’ In ‘2.3. Label-Free LC-MS/MS Quantitative Proteomics Analysis’ part.
25) L196: high concentration (1%) of
Response:
We appreciate the reviewer’s comment on language, we have already deleted ‘Carbon dioxide concentration’. ‘To investigate the effect of CO2 variations on the development of P. ostreatus PH11, young fruiting bodies of P. ostreatus PH11 were kept under a low concentration (atmospheric conditions) or high concentration (1%) of CO2.’ In ‘3.1. Phenotype of P. ostreatus under Different CO2 Concentrations’.
26) L245-L249: rewrite this part.
Response:
We appreciate the reviewer’s comment on language, we have already rewritten this part. The original part: Many DEPs are domain-containing proteins, which seem to be proteins with different biological functions. (Such as A0A067NG89, A0A067NZR1). Some are hydrolase. (Such as A0A067NG45, A0A067NTT4). Some are histone. (Such as A0A067NL48, A0A067P7W0). Some are chitinase. (Such as A0A067N5C9, A0A067NX05). Some are ribosomal protein. (Such as A0A067NG60, A0A067N2P4). In ‘3.2. Proteomic Analysis of P. ostreatus Grown under Different CO2 Conditions’ part. The modified part: A wide variety of DEPs are annotated with different biological functions. For instances, some DEPs are hydrolases (Such as A0A067NG45, A0A067NTT4), which catalyze the hydrolysis of a chemical bond. Ribosomal proteins, such as A0A067NG60, A0A067N2P4, play roles in the formation and functioning of the ribosome. Chitinases, including A0A067N5C9, A0A067NX05, play significant roles in stipe cell wall extension in mushroom. Some DEPs are histones (Including A0A067NL48, A0A067P7W0), which provide structural support for a chromosome.
27) L273-L274: why do you include this paragraph? it is unecessary. Otherwise, describe it with citation.
Response:
We really appreciate the reviewer’s critical and thoughtful comments. We are sorry for confusion. We have already deleted this paragraph: ‘Compared with those under ambient condition, the activities of soil protease, urease, amylase and phosphatase under elevated CO2 increased significantly in Pinus koraiensis.’ in ‘3.3. Gene Ontology (GO) Enrichment Analysis for DEPs’ part.
28) L280: Capitalize each word.
Response:
We really appreciate the reviewer’s critical and thoughtful comments on format, we have already modified: 3.4. Weighted Gene Co-expression Network Analysis for DEPs
29) L281: weighted gene co-expression network analysis (WGCNA)
Response:
We really appreciate the reviewer’s critical and thoughtful comments on format, we have already modified: Totally, 1333 proteins were classified into seven different modules by weighted gene co-expression network analysis (WGCNA) in ‘3.4. Weighted Gene Co-expression Network Analysis for DEPs’ part.
30) L282,L283: classified
Response:
We really appreciate the reviewer’s critical and thoughtful comments on grammar, we have already changed ‘classify’ into ‘classified’: ‘There are 602 proteins classified into turquoise module, which is the largest module. There are 374 proteins classified into blue module. There are 97 proteins classified into yellow module. There are 97 proteins classified into brown module. There are 68 proteins classified into grey module. There are 55 proteins classified into green module. There are 40 proteins classified into red module.’ in ‘3.4. Weighted Gene Co-expression Network Analysis for DEPs’ part.
31) L282-L286: rewrite this part.
Response:
We really appreciate the reviewer’s critical and thoughtful comments. We have already rewritten this part according to the sort of numbers of proteins in each module: ‘There are 602 proteins classified into turquoise module, which is the largest module. There are 374 proteins classified into blue module. There are 97 proteins classified into yellow module. There are 97 proteins classified into brown module. There are 68 proteins classified into grey module. There are 55 proteins classified into green module. There are 40 proteins classified into red module.’ in ‘3.4. Weighted Gene Co-expression Network Analysis for DEPs’ part.
Round 2
Reviewer 2 Report
L263: Totally, 1333 proteins were classified into seven different modules by weighted gene coexpression network analysis (WGCNA) (602 proteins in the largest turquoise module, 374 in blue, 97 in yellow, 97 brown, 68 in grey, 55 green, and 40 in red module). Eigengene of each module in each treatment show different expression level of gene in each module. Proteins in MEblue module show relative high expression level in HCO compared with LCO, whereas proteins in MEturquoise module show relative high expression level in LCO compared with HCO (Table 1, Figure 6,9).
L358-L436: These paragraphs and sentences are simple listing of protein names. It is not the way of writing discussion. It needs compact and concise descriptions and discussion on your findings.
L441 and L461: Are they subtitles? Why do the words have capital letter?
Author Response
Responses to the comments:
1) L263: Totally, 1333 proteins were classified into seven different modules by weighted gene coexpression network analysis (WGCNA) (602 proteins in the largest turquoise module, 374 in blue, 97 in yellow, 97 brown, 68 in grey, 55 green, and 40 in red module). Eigengene of each module in each treatment show different expression level of gene in each module. Proteins in MEblue module show relative high expression level in HCO compared with LCO, whereas proteins in MEturquoise module show relative high expression level in LCO compared with HCO (Table 1, Figure 6,9).
Response:
We really appreciate the reviewer’s critical and thoughtful comments on language. We have already rewritten this paragraph according to reviewer’s suggestion, which made the language of paragraph concise and easy to understand:
Totally, 1333 proteins were classified into seven different modules by weighted gene coexpression network analysis (WGCNA) (602 proteins in the largest turquoise module, 374 in blue, 97 in yellow, 97 brown, 68 in grey, 55 green, and 40 in red module).
2) L358-L436: These paragraphs and sentences are simple listing of protein names. It is not the way of writing discussion. It needs compact and concise descriptions and discussion on your findings.
Response:
We really appreciate the reviewer’s comments. The discussion part and conclusion part were re-written completely.
3) L441 and L461: Are they subtitles? Why do the words have capital letter?
Response:
We appreciate the reviewer’s comment on format. These sentences have been deleted form the revised manuscript.